**Data Availability Statement:** All relevant data are within the paper.

**Funding:** This study was supported by grants from: Sao Paulo Research Foundation (FAPESP

# Shorter telomere length and suicidal ideation in familial bipolar disorder

Daniela Martinez[1,2,3], Catharina Lavebratt[2,4], Vincent Millischer[2,4], Vanessa de Jesus R. de Paula[1], Thiago Pires[3], Leandro Michelon[1], Caroline Camilo[1], Nubia Esteban[3], Alexandre Pereira[3], Martin Schalling[1,2,4], Homero Vallada[1,2,4]*

1 Departamento & Instituto de Psiquiatria, Faculdade de Medicina, Universidade de Sao Paulo, Sao Paulo, SP, Brazil, 2 Center for Molecular Medicine, Karolinska University Hospital, Stockholm, Sweden, 3 Laboratório de Genética e Cardiologia Molecular, Instituto do Coração, Faculdade de Medicina FMUSP, Universidade de Sao Paulo, Sao Paulo, SP, Brazil, 4 Department of Molecular Medicine and Surgery, Karolinska Institutet, Stockholm, Sweden

* hvallada@usp.br

## Abstract

Bipolar Disorder (BD) has recently been related to a process of accelerated aging, with shortened leukocyte telomere length (LTL) in this population. It has also been observed that the suicide rate in BD patients is higher than in the general population, and more recently the telomere length variation has been described as shorter in suicide completers compared with control subjects. **Objectives** The aim of the present study was to investigate if there is an association between LTL and BD in families where two or more members have BD including clinical symptomatology variables, along with suicide behavior. **Methods** Telomere length and single copy gene ratio (T/S ratio) was measured using quantitative polymerase chain reaction in a sample of 143 relatives from 22 families, of which 60 had BD. The statistical analysis was performed with a polygenic mixed model. **Results** LTL was associated with suicidal ideation (p = 0.02) as that there is an interaction between suicidal ideation and course of the disorder (p = 0.02). The estimated heritability for LTL in these families was 0.68. In addition, covariates that relate to severity of disease, i.e. suicidal ideation and course of the disorder, showed an association with shorter LTL in BD patients. No difference in LTL between BD patients and healthy relatives was observed. **Conclusion** LTL are shorter in subjects with familial BD suggesting that stress related sub-phenotypes possibly accelerate the process of cellular aging and correlate with disease severity and suicidal ideation.

## Introduction

Bipolar disorder (BD) is a debilitating and chronic illness, with an important genetic component in its etiopathogenesis. Clinically, BD is characterized by recurrent episodes of mania or hypomania and depression [1], followed by physiological changes and accelerated biological aging processes [2–6]. First episodes of bipolar disorder usually emerge during adolescence or young adulthood and can be incapacitating for long periods of the individual's life [7]. In

grant #2015/14614-6 to DM), CAPES
(Coordenação de Aperfeiçoamento Pessoal de
Nível Superior) 88887.463672/2019-00, Brazilian
Federal Agency for Higher Education (CAPES
PROEX #1229245 to CC & #1531878 to LM),
Brazilian Research Council of research (CNPq grant
#448735/2014-8), Swedish Research Council
(grants 2013-6652, 2016-02653) and funds from
the Karolinska Institutet and Karolinska University
Hospital.

**Competing interests:** The authors have declared
that no competing interests exist.

addition, BD patients are at an increased risk for ideation, attempts, and completed suicide, which can be 10 to 30 times more prevalent when compared with the general population [8, 9].

Bipolar disorder patients present neurobiological processes and substrates similar to aging, such as brain volume reduction [10–12]. Brian Hallahan et al. [10], showed 321 individuals with BD type I had increased right lateral ventricular, left temporal lobe, and right putamen volumes. The immunological age, and immunosenescence in BD has been associated with an increased proportion of late differentiated T cells (CD3+CD8+CD28-CD27) in peripheral blood mononuclear cells (44 male and female euthymic patients with BD type I; β = 0.360, p = .013) [13]. Oxidative stress levels were higher in BD-mania for TBARS (P < .0001) and uric acid (P < .0001); in BD-depression for TBARS (P = .02); and BD-euthymia for uric acid (P = .03) and telomere shortening in a meta-analysis with forty-four studies (n = 3,767: BD = 1,979; HCs = 1,788) [14]. In addition, these patients more frequently present other medical conditions during the illness, such as cardiovascular and endocrine diseases and even dementia, leading some researchers to consider BD as a disorder of accelerated aging [15, 16].

A "molecular clock", a figurative term related to cellular aging has been associated with progressive telomere shortening, i.e., telomeres shorten progressively with age, inflammation, and oxidative stress [3, 17]. Moreover, a meta-analysis with twenty-four studies (43,725 participants) indicated that shorter telomere length is associated with risk of cardiovascular disease [18]. Furthermore, shorter telomere length was associated with infection-related diseases in 75,309 individuals randomly invited from Danish Civil Registration System (95% confidence interval) [19]. Subsequently, leukocyte telomere length (LTL) was associated with cellular senescence and longevity, as well as with disorders associated with aging [20]. In addition to the variation in LTL with age (aged 19–64 years at baseline and follow-up of 12 years), studies have recently reported heritability as a very important contributory factor to the variation in human LTL estimated at 64% (95% CI 39% to 83%) with 22% (95% CI 6% to 49%) of shared environmental effects in 355 monozygotic and 297 dizygotic (same- sex twins). Heritability of age-dependent LTL attrition rate was estimated at 28% (95% CI 16% to 44%) [20]. A meta-analysis of telomere length studies, with a total of 19,713 participants, showed high heritability for the LTL phenotype, estimated between 34–82% (95% CI 0.64–0.76) [21]. Another meta-analysis investigation, including 14,827 participants, reported LTL shortening in posttraumatic stress disorder (PTSD), anxiety disorders, depressive disorders, BD, and psychosis (Hedge's g = −0.50, p< 0.001) [22]. With regard specifically to BD, both LTL reduction [21, 22], and longer LTL [23] have been observed. However, the longer LTL is interpreted as an effect of lithium acting as a neuroprotective drug during the treatment of BD type 1 or 2 (n = 256) and healthy controls (n = 139), and BD had 35% longer telomeres compared with controls (P<0.0005) [23, 24]. In another study 200 patients with BD had longer LTL, positively correlated with lithium treatment in patients treated for more than two years (p = 0.037) [24].

The biological function of telomeric length in leukocytes of BD patients is little explored in the literature, and as a consequence, it is still unclear whether there is a direct or indirect relationship with the severity of the disease. Leukocyte telomere length is therefore considered a promising biomarker of biological aging and accelerated aging and leukocytes had the same rate compared with skin cells, fat cells, and muscle tissues in a cohort of 87 subjects, being a peripheral blood sample more reliable with the rest of the human body [25]; Therefore, in the present study, we investigated LTL in affected and unaffected BD family members and examined the relationship between LTL and the effect of the disease course and severity of BD clinical symptoms.

## Materials and methods

### Subjects

This was a cross-sectional study using 22 families (143 individuals) with two or more individuals per family affected by BD totaling 60 individuals diagnosed with BD. DNA samples of these Family members are stored in a DNA bank from the Instituto de Psiquiatria do Hospital das Clínicas da Faculdade de Medicina da Universidade de São Paulo.

A brief description of the selection and assessment of families is presented below, but further details can be seen elsewhere [17, 26]. The families containing several members with BD were initially identified using the "Families Study" method [5, 27], which gathers information through interviews with each potential participant. Subsequently, each family member was assessed using the Schedule for Affective Disorders and Schizophrenia–Lifetime version (SADS-L) [28] and the OPCRIT (Operational Criteria Checklist for Psychotic Illness and Affective Illness) software (version 3.3), an operational criteria checklist for psychotic illness which provides a simple and reliable method of applying multiple operational diagnostic criteria in psychosis studies [29]. Complementary information from medical records (such as diagnosis, medications, response to treatment, etc.) was also collected when available.

It is important to emphasize that a minority of the family members were affected by other psychiatric disorders such as major depressive disorder (MDD), minor depression, schizophrenia, intermittent depressive disorder, alcoholism, hypomania, and dementia, as characterized in Table 1.

This study was approved by the local research ethics committee (CEP–FMUSP process number #215/15).

### Clinical variables–OPCRIT guidelines and ratings

All the clinical variables included in the present study were extracted from the OPCRIT software, version 3.3 [29]. A total of five clinical symptoms were included in the analysis: (1) duration of disease, (2) number of hospitalizations, (3) suicidal ideation, (4) course of disorder, and (5) psychiatric comorbidities. A brief description of two of these variables according to the Guidelines & Ratings of the OPCRIT follows below. Suicidal ideation is classified as "preoccupation with thoughts of death", "thinking of suicide", "wishing to be dead", and "attempts to kill self". The clinician allocates a score from 0–3; score '0' if absent, score '1' for duration of at least one week, '2' for at least two weeks duration, and '3' for at least one month". The course of disorder was scored in a hierarchical fashion; score '1' for a single episode with good recovery, '2' for multiple episodes with good recovery between episodes, '3' for multiple episodes with partial recovery between episodes, '4' for continuous chronic illness, and '5' for continuous chronic illness with deterioration" [29].

### DNA extraction

Genomic DNA extraction was performed using a phenol-chloroform standard protocol [30] and stored in a DNA bank. DNA samples were maintained at -20˚C and reassessment of the DNA quality was performed using NanoDrop and agarose gel electrophoresis.

### LTL measurement

The leukocyte telomere length (LTL) was determined using real-time quantitative PCR (qPCR) according to the protocol of Cawthon et al. [31] where the relative telomere to single copy gene (T/S) ratio was determined using a standard curve. In brief, each DNA sample (4.0 ng) was assessed for the telomere and the single-copy gene (hemoglobin-b, HBB) in triplicate

**Table 1. Clinical assessment of the 143 participants from the 22 selected families.**

| Variables | Men N = 62 | Women N = 81 | Total, N (%) N = 143 |
|---|---|---|---|
| **Age (years)** | | | |
| Average | 47 | 44 | 45 |
| Median | 43 | 40 | 40 |
| Standard-deviation (SD) | 18 | 14 | 16 |
| **Status of Disease** | | | |
| Healthy | 26 | 22 | 48 (33.6) |
| Bipolar Disorder | 26 | 34 | 60 (42) |
| Major Depression | 4 | 13 | 17 (11.9) |
| Minor Depression | 2 | 3 | 5 (3.5) |
| Schizophrenia | 2 | 3 | 5 (3.5) |
| Intermittent Depressive Disorder | 0 | 4 | 4 (2.8) |
| Alcoholism | 1 | 1 | 2 (1.4) |
| Hypomania | 0 | 1 | 1 (0.7) |
| Dementia | 1 | 0 | 1 (0.7) |
| **Psychiatric comorbidities*** | | | |
| No | 17 | 34 | 51 (53.7) |
| Yes | 19 | 25 | 44 (46.3) |
| **Suicidal Ideation** | | | |
| No | 29 | 42 | 71 (64.0) |
| At least 1 week or suicide attempt | 10 | 14 | 24 (21.6) |
| At least 2 weeks | 1 | 2 | 3 (2.7) |
| At least 1 month | 4 | 9 | 13 (11.7) |
| **Course of disorder** | | | |
| Single episode w/ good recovery | 6 | 6 | 12 (13.3) |
| Multiple episodes w/ good recovery between | 12 | 25 | 37 (41.1) |
| Multiple episodes w/ partial recovery between | 8 | 15 | 23 (25.6) |
| Continuous chronic illness | 4 | 10 | 14 (15.6) |
| Continuous chronic illness w/ deterioration | 4 | 0 | 4 (4.4) |
| **Number of hospitalization** | | | |
| Average | 4.39 | 5.06 | 4.79 |
| Median | 2 | 2 | 2 |
| SD | 7.56 | 7.69 | 7.60 |
| **Duration of disease (years)** | | | |
| Average | 16.91 | 18.00 | 17.60 |
| Median | 17 | 17 | 17 |
| SD | 10.45 | 11.70 | 11.21 |
| **T/S ratio** | | | |
| Average | 1.36 | 1.40 | 1.38 |
| Median | 1.36 | 1.35 | 1.36 |
| SD | 0.33 | 0.33 | 0.33 |
| **Total, N (%)** | 62 (43.4) | 81(56.6) | 143 (100) |

*Main comorbidities were Depression, Anxiety Disorders, Alcohol and Drug abuse.

within the same 384-well plate, amplified using Power SYBR Green in 10 μl total reaction volume. The reaction was performed on QuantStudio 7 Flex (Applied Biosystems; Life Technologies, Carlsbad, CA, USA) with the following conditions: 95˚C for 10 min, followed by 39

repeats of 95˚C for 15 s and 60˚C for 1 min, followed by a dissociation stage to monitor amplification specificity. The same standard curve of pooled DNA from these patient samples ranging from 10 ng to 0.016 ng was run on each plate for both genes and used to determine the quantity of each gene for each sample. This allowed control of the differences in the efficiencies between the Telomere and HBB. The gene quantities were then used to determine the T/S ratio for each sample. DNA samples with a Ct (cycle threshold) standard deviation of ≥ 0.4 between triplicates or a Ct value outside the standard curve were omitted from the analyses. The correlation coefficients of the standard curves were above 0.99 for each primer set and 384-plate. The primers (10 nM) for the telomere PCR were Tel1 (5'- GGTTTTTGAGGGTG AGGGTGAGGGTGAGGGTGAGGGT-3') and Tel2 (5'- TCCCGACTATCCCTATCCCTATC CCTATCCCTATCCCTA-3') and for the HBB PCR were HBB1 (forward primer; 5'-GCTT CTGACACAACTGTGTTCACTAGC-3') and HBB2 (reverse primer; 5'-CACCAACTTCATC CACGTTCACC-3'). At the end of each reaction, the dissociation curve was performed (melting curve) to verify the specificity of the reaction. For each reaction, three inter-plate calibrators (control samples) were run in each plate.

## Statistical analysis

All the LTL-covariate association tests and heritability calculations were performed using SOLAR software, version 6.6.2 and R version 3.2.0 software with the Coxme package using the lmekin function. For this calculation a polygenic mixed model was used (Mathematical and Statistical Methods for Genetic Analysis | Kenneth Lange | Springer) which is based on genetic decomposition $y_i$, phenotype. It is described as a linear function given by $y_i = \mu + \beta_j X_{;ij} + g_i + e_i$, where $\mu$ is the global average trace and $X_{;ij}$ and $\beta_j$ represent, respectively, the design matrix and the vector of parameters associated with environmental fixed effects (covariates), $g_i$ is the random additive genetic effect, and $e_i$ is the residual random environmental effect. This model also estimates the effects of each covariate included ($\beta_j$), and the respective statistical significance, by the Wald test. The relationship structure was obtained through the pedigrees of this study that were used to estimate the relationship matrices (kinship matrix: $X_{;ij}$) and variance and covariance matrices [32]. Thus, all the calculations were adjusted for the family component. The total phenotype variance ($y^2 + y^2$) is the result of the sum of the polygenic variance ($y^2$), attributed to genetic effects, and the environmental variance ($y^2$), assigned to the residual effect [33]. Thus, the heritability ($h^2$) was calculated as the ratio of the total phenotypic variation attributed to the genetic effect:

$$h^2 = \frac{\gamma_g^2}{\gamma_g^2 + \gamma_e^2}$$

## Results

### Descriptive analysis

In total, 143 individuals from 22 families were analyzed, of which 95 had a psychiatric diagnosis. The most prevalent illness was BD, corresponding to 42% of the total sample (Table 1). Men and women did not differ significantly in age and the combined mean age was 40 years, indicating that the studied population is mostly composed of middle-aged adults. The data was available at the public repository Figshare (https://doi.org/10.6084/m9.figshare.20353203.v1).

The T/S ratio was normally distributed (D = 0.05, p = 0.83). The quantile-quantile plot of the T/S ratio shows a single point outside the confidence interval, which did not interfere largely with the normality of data (Fig 1). Males and females did not differ significantly in telomere length (p = 0.41).

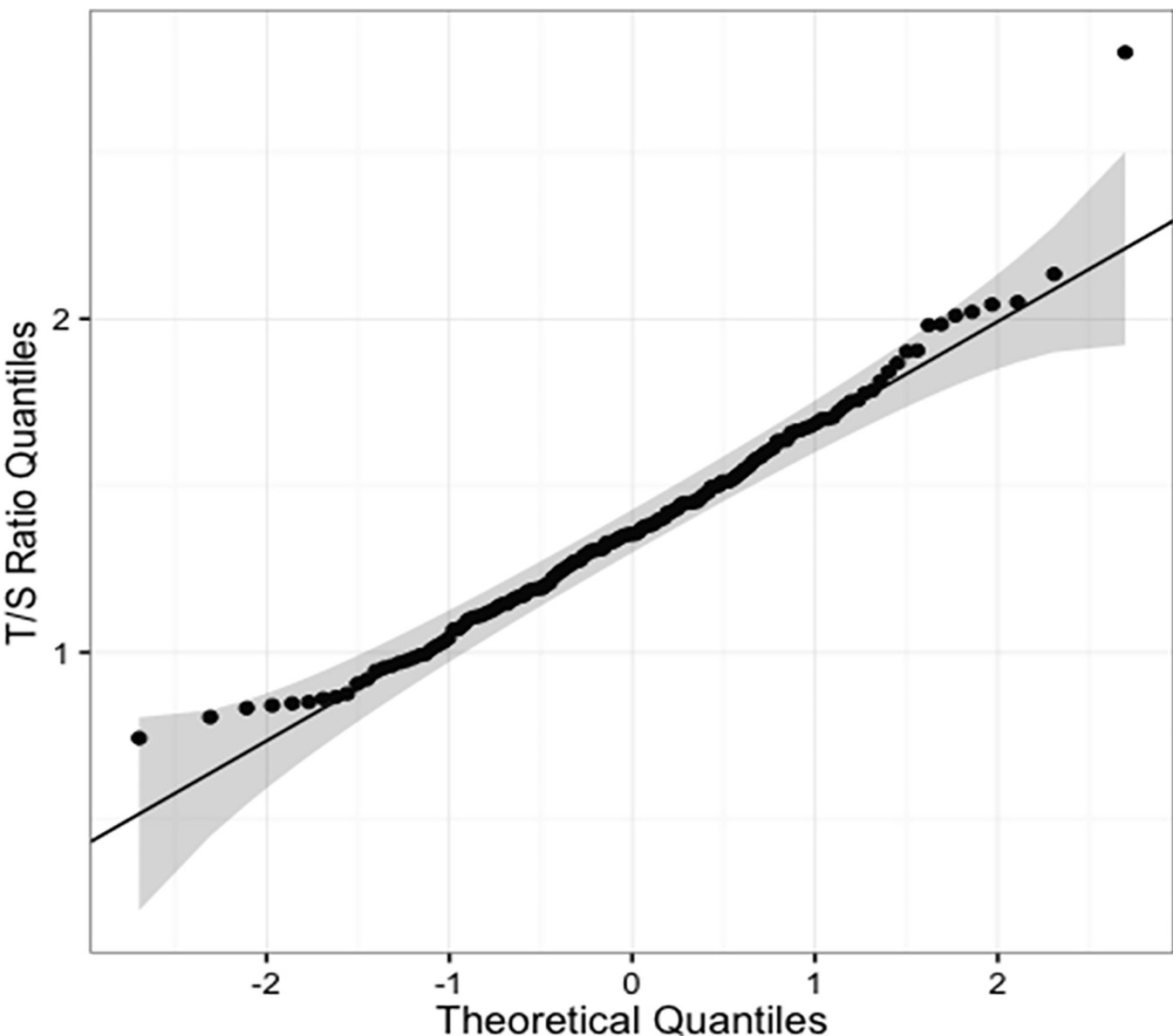

**Fig 1. Quantile-quantile plot (Q-Q plot) of T/S ratio.** The Q-Q plot shows normality in the distribution of T/S ratio data (D = 0.05, p = 0.83). D = Kolmogorov-Smirnov.

### LTL and BD association

There was no correlation between BD and LTL using a polygenic mixed model, adjusted for sex and age (β = 0.02, p = 0.66) or with individuals with other psychiatric disorders (PD) (Fig 2).

### LTL and clinical covariates

We tested for associations between LTL and all clinical covariates of the individuals. Only two showed a significant association with the T/S ratio, including suicidal ideation (β = -0.06; p = 0.02) and interaction between suicidal ideation and course of disorder (β = -0.03; p = 0.02). Fig 3 presents telomere length measurements according to severity of suicidal ideation (no periods, at least one week, two or four weeks of suicidal ideation). There was a significant

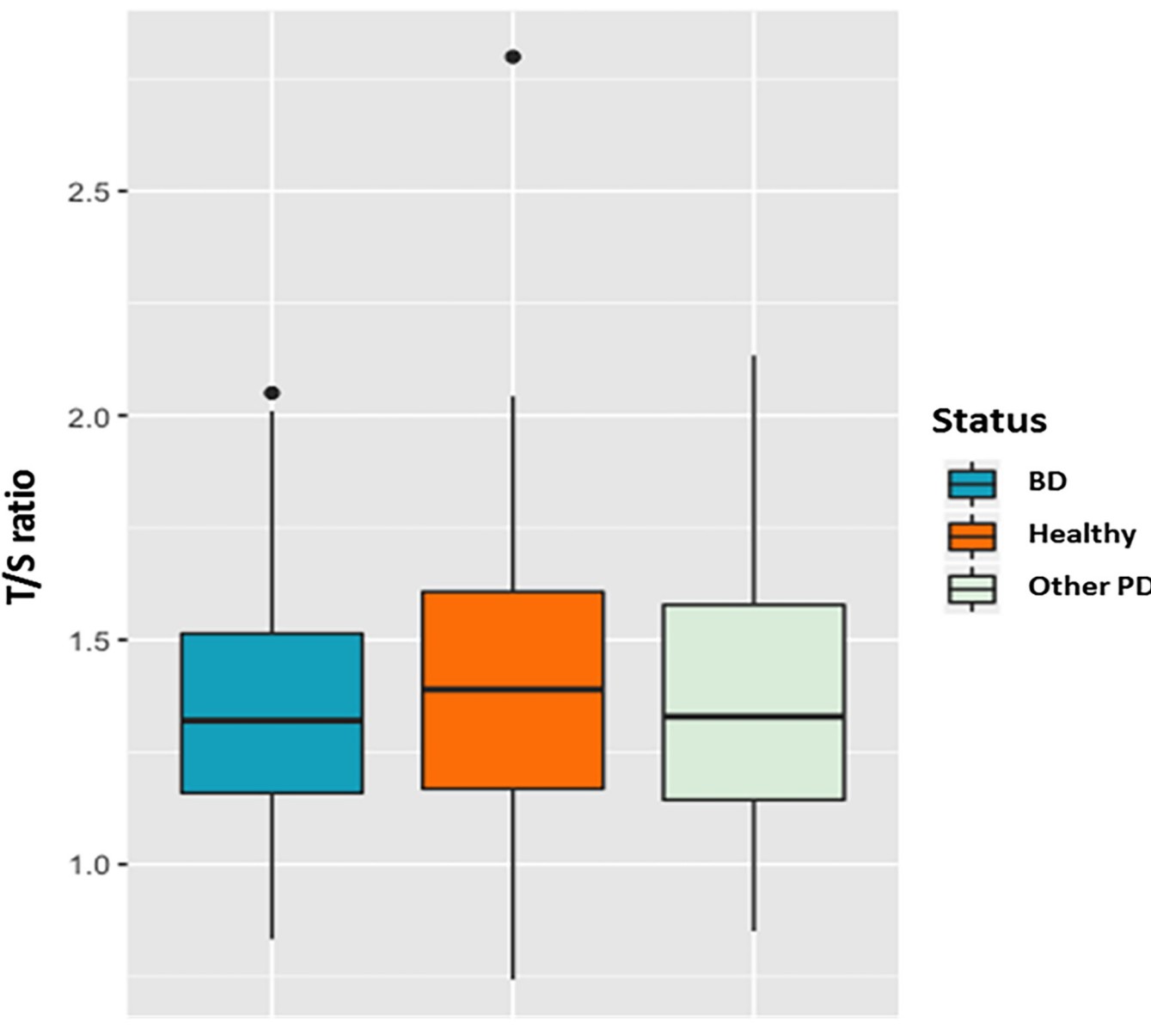

**Fig 2. Boxplot of the leukocyte telomere length measure (T/S ratio) for bipolar patients (BD), the healthy group, and patients with other psychiatric disorders (PD).** No correlation between groups (β = 0.02, p = 0.66).

difference between the groups (p = 0.02) and the estimate of the effect associated with suicidal ideation (β = -0.06) indicates that the longer the period of suicidal ideation, the shorter the telomere length. Number of hospitalizations, duration of disease, and comorbidity did not correlate with LTL.

## Heritability of LTL

The heritability of telomere length was also calculated, and its overall estimate, adjusted for sex and age was 0.68 (Table 2). The heritability was not affected by the addition of the BD dichotomous variable to the analysis (p>0.05; βBD = 0.02).

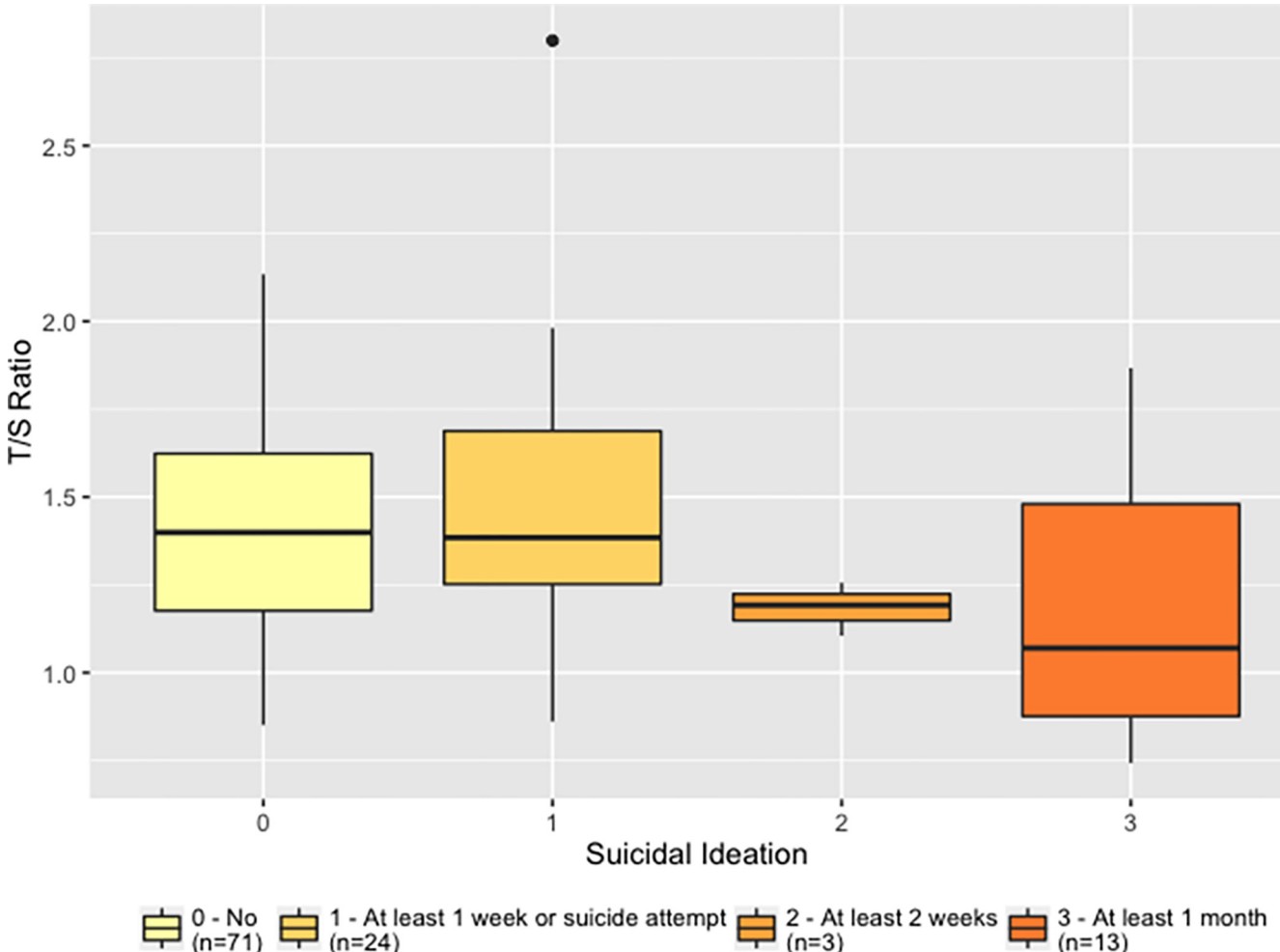

**Fig 3. The leukocyte telomere length (LTL) measure (T/S ratio) in relation to suicidal ideation.** LTL was associated with the variable "suicidal ideation" ($\beta$ = -0.06; p = 0.02). *0 = No episode of suicidal ideation; 1 = At least 1 week of suicidal ideation or suicide attempt; 2 = At least 2 weeks of suicidal ideation; 3 = At least 1 month.*

## Discussion

In order to further explore the biological function of telomeric length in leukocytes of BD patients and its relationship with the severity of the disease, we investigated LTL in affected and unaffected BD members from 22 families and examined the correlation between LTL and the effect of the disease course and BD clinical symptoms. A statistically significant association was observed between LTL and suicidal ideation (p<0.05) and the interaction between suicidal

**Table 2. Heritability of LTL.** Calculation of heritability of the leukocyte telomere length (LTL) in 22 Bipolar families.

| Models | B values | P values | H$^2$ |
|---|---|---|---|
| Null | | | 0.58 |
| Sex (female) | 0.04 | 0.43 | 0.58 |
| Age | 0.00 | 0.01 | 0.68 |
| Sex + Age | B$_{sex}$ = 0.02/$\beta_{age}$ = 0.00 | B$_{sex}$ = 0.63/$\beta_{age}$ = 0.01 | 0.68 |
| Sexn+ Age + Bipolar | B$_{sex}$ = 0.02/$\beta_{age}$ = 0.00/$\beta_{BD}$ = 0.02 | >0.05 | 0.64 |

ideation and course of disorder (p<0.05). However, using a polygenic mixed model, the BD phenotype did not correlate significantly with LTL variation.

It is already well reported that bipolar disorder is associated with a high risk of suicide attempts and suicide death [34], however there are very few studies associating shortening telomeres with suicide death in BD patients. In 2017, Otsuka et al. reported an association between shorter telomeres and mitochondrial DNA copy number alterations in post-mortem dorsolateral prefrontal cortex of 508 suicide completers when compared with 535 controls [35]. The authors suggest that the duration of exposure to a psychological stressor, culminating in suicide may contribute to the shortening effect. More recently, Kim et al. demonstrated that suicide completers had a significantly shorter LTL compared with healthy controls [36]. The underlying mechanisms between suicide and telomere shortening remain to be elucidated.

Data from the Utah Suicide Genetic Risk Study (USGRS) biobank showed that among the sample of 2,672 suicide deaths, compared to non-suicide deaths, both PER1 (Period Circadian Regulator 1) and SNAPC1 (Small Nuclear RNA Activating Complex Polypeptide 1) genes show evidence of suicide risk in bipolar disorder and schizophrenia [37]. The PER1 protein is important for the maintenance of circadian rhythms in cells, and its deficiency in mice causes loss of rhythmic telomerase activities, TERT mRNA oscillation, and shortened telomere length [11, 38].

Other studies highlighted the importance of oxidative stress for the telomere shortening process [39, 40]. Furthermore, these studies suggest that an attenuated oxidative stress defense and telomerase deficiency contribute to telomere shortening in oligodendrocytes in MDD [41]. In an animal model study, the authors found an association between stress and telomere length in mice and also an association between mice injected with corticosterone and shortened telomeres [41]. Since stress is associated with suicidal ideation and shortened telomeres it is possible that our findings are attributed to changes in hormonal responses as a consequence of stress.

With regard specifically to BD, associations with LTL reduction [42, 43], and longer LTL [23, 24] have been reported. However, the longer LTL is interpreted as an effect of lithium acting as a neuroprotective drug during the treatment of BD [23, 24]. Only one study reported on telomere length in BD in relation to a demographically controlled reference sample, and the authors found no association between LTL and BD [44]. The first report investigating mood disorders and LTL found an association between shorter LTL and depression or BD when compared to controls [15]. More recently, Powell et al. [45] provided evidence that a shortened telomere length is associated with family BD risk. The telomeric length of control subjects and patients with BD and their respective relatives was compared and it was found that both the BD group and the BD relative group presented smaller telomeric sizes in relation to the control groups. Other subsequent studies investigating LTL and BD did not confirm an overall association between BD and shorter LTL when compared to healthy controls [15, 16, 42, 46–48]. Even so, no differences were found in telomere length between BD and healthy relatives in 22 families.

An interesting approach to further investigate the association between BD and LTL includes the analysis of the effect of lithium, one of the most effective treatments for BD patients. Martinsson et al. showed that BD patients who responded well to lithium treatment had longer telomeres compared with BD patients who were poor responders to lithium and healthy controls [23]. One possible mechanism of lithium, which has a neuroprotective effect, could be to affect telomere biology through its inhibitory effect onGSK3beta, thus reducing the telomere-shortening process [23, 24, 44]. A recent review shows how lithium treatment should be implicated in the pathophysiology of BD, especially in mitochondrial dysfunction and telomere shortening [49]. In our study, none of the BD patients received lithium treatment and

therefore no comparison was possible. Although we did not find a correlation between BD phenotype and LTL variation, we observed an association between shorter LTL and two clinical phenotypes in BD: "suicidal ideation" and an interaction between "suicidal ideation" and "course of disorder", which indicate the severity of BD. This result suggests that the longer the period of suicidal ideation (at least 1 month of suicidal ideation–see Fig 3) and the course of the disorder, the shorter the telomere length.

In addition, a high estimated heritability of LTL was observed in these BD families ($h^2 =$ 0.68). In the present study we used familial samples in an attempt to reduce confounders based on differences between groups such as socioeconomic status, stress, and educational level. Epidemiological studies have shown telomere length as a complex heritable trait with estimated heritability derived from twin studies of from 36% to 82% compared to 34% to 50% from familial studies [50]. Slagboom & Droog in 1994 were the first to report an estimate of the heritability of LTL at 0.78 from 115 twin pairs [51]. Subsequent studies confirmed the heritability of LTL in twins and triplets [15, 52], but also in family studies [53–57]. Broer et al. showed, in a meta-analysis with a total of 19,713 subjects, large variation in the heritability of LTL, ranging between 0.34 and 0.82 [21]. More recently, Kim J and collaborator [58] showed high heritability of telomere length across three generations of Korean families, 287 individuals in total, with no differences between paternal and maternal telomere length. In another study with mothers and newborns, LTL was more strongly correlated with the mother's LTL [White (56%) and Hispanic (29%)], than with the father's LTL, showing ethnic and sex differences in the telomere length variation [59]. The variation in LTL heritability between studies may be attributed to family structure, age of the participants, and technical variations. Previous studies have also shown that LTL varies between individuals in a given age group and some have suggested that LTL could be a marker of biological aging [60].

Our findings need to be considered in the light of some methodological limitations including: a relatively small sample size and validity of the clinical information collected in some individuals.

In conclusion, our investigation observed an association of a shorter LTL with suicidal ideation in subjects with familial BD and a correlation of a shorter LTL and interaction between suicidal ideation and course of illness. These results reinforce the hypothesis of stress (oxidative and/or psychological) possibly interacting with telomere length. Future investigations using independent BD families are important to confirm the present findings.

## Acknowledgments

The authors are thankful to Dr. Helena Brentani and her group for their thoughtful comments and suggestions. D.M. is grateful for the members of the genetic laboratories for their technical assistance.

## Author Contributions

**Conceptualization:** Catharina Lavebratt, Leandro Michelon, Caroline Camilo, Nubia Esteban, Alexandre Pereira, Martin Schalling, Homero Vallada.

**Data curation:** Daniela Martinez, Catharina Lavebratt, Vincent Millischer, Vanessa de Jesus R. de Paula, Leandro Michelon, Nubia Esteban, Alexandre Pereira, Martin Schalling.

**Formal analysis:** Daniela Martinez, Vanessa de Jesus R. de Paula, Leandro Michelon, Nubia Esteban, Alexandre Pereira, Homero Vallada.

**Funding acquisition:** Catharina Lavebratt, Martin Schalling, Homero Vallada.

**Investigation:** Catharina Lavebratt, Nubia Esteban, Martin Schalling, Homero Vallada.

**Methodology:** Daniela Martinez, Catharina Lavebratt, Vincent Millischer, Thiago Pires, Caroline Camilo, Alexandre Pereira, Martin Schalling, Homero Vallada.

**Project administration:** Daniela Martinez, Catharina Lavebratt.

**Resources:** Catharina Lavebratt, Thiago Pires, Martin Schalling, Homero Vallada.

**Software:** Thiago Pires, Martin Schalling.

**Supervision:** Martin Schalling, Homero Vallada.

**Validation:** Daniela Martinez, Vincent Millischer, Thiago Pires, Caroline Camilo, Martin Schalling, Homero Vallada.

**Visualization:** Daniela Martinez, Thiago Pires.

**Writing – original draft:** Daniela Martinez, Vincent Millischer, Vanessa de Jesus R. de Paula, Thiago Pires, Leandro Michelon, Caroline Camilo, Nubia Esteban, Alexandre Pereira, Martin Schalling, Homero Vallada.

**Writing – review & editing:** Daniela Martinez, Martin Schalling, Homero Vallada.

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
