## [Decision Letter · Decision Letter 0]

14 Jun 2022

PONE-D-22-12118Shorter telomere length and suicidal ideation in familial bipolar disorderPLOS ONE

Dear Dr. Vallada,

Thank you for submitting your manuscript to PLOS ONE. After careful consideration, we feel that it has merit but does not fully meet PLOS ONE’s publication criteria as it currently stands. Therefore, we invite you to submit a revised version of the manuscript that addresses the points raised during the review process.

We look forward to receiving your revised manuscript.

Kind regards,

Vincenzo De Luca

Academic Editor

PLOS ONE

Journal Requirements:

"This study was supported by grants from: Sao Paulo Research Foundation (FAPESP grant #2015/14614-6 to DM), CAPES (Coordenação de Aperfeiçoamento Pessoal de Nível Superior) 88887.463672/2019-00, Brazilian Federal Agency for Higher Education (CAPES PROEX #1229245 to CC & #1531878 to LM), Brazilian Research Council of research (CNPq grant #448735/2014-8), Swedish Research Council (grants 2013-6652, 2016-02653) and funds from the Karolinska Institutet and Karolinska University Hospital. The funders had no role in the study design, data collection and analysis, decision to publish, or the preparation of the manuscript."

"This study was supported by grants from: Sao Paulo Research Foundation (FAPESP grant #2015/14614-6 to DM), CAPES (Coordenação de Aperfeiçoamento Pessoal de Nível Superior) 88887.463672/2019-00, Brazilian Federal Agency for Higher Education (CAPES PROEX #1229245 to CC & #1531878 to LM), Brazilian Research Council of research (CNPq grant #448735/2014-8), Swedish Research Council (grants 2013-6652, 2016-02653) and funds from the Karolinska Institutet and Karolinska University Hospital. "

Reviewers' comments:

Reviewer's Responses to Questions

**Comments to the Author**

1. Is the manuscript technically sound, and do the data support the conclusions?

Reviewer #1: Yes

2. Has the statistical analysis been performed appropriately and rigorously? 

Reviewer #1: Yes

3. Have the authors made all data underlying the findings in their manuscript fully available?

Reviewer #1: Yes

4. Is the manuscript presented in an intelligible fashion and written in standard English?

Reviewer #1: Yes

5. Review Comments to the Author

Reviewer #1: Introduction

A fascinating and much needed study.

You wrote: “Bipolar disorder patients present neurobiological processes and substrates similar to aging, such as brain volume reduction (7–9); cognitive impairment (Aprahamian et al., 2013); increase in circulating CD8+CD28- cells (11); elevated nitric oxide levels (3,12); and telomere shortening (13). In addition, these patients more frequently present other medical conditions during the course of the illness, such as cardiovascular and endocrine diseases and even dementia, leading some researchers to consider BD as a disorder of accelerated aging (14,15”

[This a key statement for your hypothesis. Can you please present sample sizes, study design (prospective…cohorts) attrition, and numerical and statistical outcomes?] As written we have no idea of the risk of bias in the sample or strength of association]

You wrote: “Moreover, epidemiological studies have shown a correlation between telomere shortening and increased mortality due to cardiovascular and infection-related diseases (1,17). Subsequently, leukocyte telomere length (LTL) was associated with cellular senescence and longevity, as well as with disorders associated with aging (18). In addition to the variation in LTL with age, studies have recently reported heritability as a very important contributory factor to the variation in human LTL (19). A meta-analysis of telomere length studies showed high heritability for the LTL phenotype, estimated between 34-82% (20).”

[Again, can you please present sample sizes, study design (prospective…cohorts) attrition, and numerical and statistical outcomes?] As written we have no idea of the risk of bias in the sample or strength of association].

“Another meta-analysis investigation reported LTL shortening in posttraumatic stress disorder (PTSD), anxiety disorders, depressive disorders, BD, and psychosis (21). With regard specifically to BD, both LTL reduction (22,23), and longer LTL (24,25) have been observed. However, the longer LTL is interpreted as an effect of lithium acting as a neuroprotective drug during the treatment of BD (24,25).”

[Same comment]

You wrote: “ The biological function of telomeric length in leukocytes of BD patients is little explored in the literature,”

[Please explain why you did not study telomere length in other cell types. Clearly leucocytes are related to your inflammation hypothesis. Can you please be explicit?

Sample:

You wrote” This was a cross-sectional study using 22 families with two or more individuals per family affected by BD”

[how many BD patients are there in your medical system? How were these patients chosen? (for another study?) How does this affect generalisability?]

Results:

You wrote: “major depressive disorder (MDD), minor depression, schizophrenia, intermittent depressive disorder, alcoholism, hypomania, and dementia,”

[How do the comorbidities affect your results?]

You wrote: The same standard curve of pooled DNA from these patient samples ranging from 10 ng to 0.016 ng was run on each plate for both genes and used to determine the quantity of each gene for each sample. This allowed control of the differences in the efficiencies between the Telomere and HBB.”

[how did the differences in DNA sample size affect analysis reliability?]

You wrote; “Figure 3 presents telomere length measurements according to severity of suicidal ideation (no periods, at least one week, two or four weeks of suicidal ideation). There was a significant difference between the groups (p = 0.02) and the estimate of the effect associated with suicidal ideation (β = - 0.06) indicates that the longer the period of suicidal ideation, the shorter the telomere length.”

[This is an important outcome and is an important criterion in measuring strength of association according to Hill’s criteria]

Overall a great study.

6. PLOS authors have the option to publish the peer review history of their article (what does this mean?). If published, this will include your full peer review and any attached files.

Reviewer #1: **Yes: **Roger E. Thomas

---

## [Author Response · Author response to Decision Letter 0]

22 Sep 2022

Reviewer #1:

Introduction

A fascinating and much needed study.

1. You wrote: “Bipolar disorder patients present neurobiological processes and substrates similar to aging, such as brain volume reduction (7–9); cognitive impairment (Aprahamian et al., 2013); increase in circulating CD8+CD28- cells (11); elevated nitric oxide levels (3,12); and telomere shortening (13). In addition, these patients more frequently present other medical conditions during the course of the illness, such as cardiovascular and endocrine diseases and even dementia, leading some researchers to consider BD as a disorder of accelerated aging (14,15”). [This a key statement for your hypothesis. Can you please present sample sizes, study design (prospective…cohorts) attrition, and numerical and statistical outcomes?] As written we have no idea of the risk of bias in the sample or strength of association]

We appreciate the care taken with the reading and the suggestions, which indeed enriched the manuscript. The following changes were added to the text, which is underlined for ease of reading, on page 4:

“Bipolar disorder patients present neurobiological processes and substrates similar to aging, such as brain volume reduction [10–12]. Brian Hallahan et al [10], showed 321 individuals with BD type I had increased right lateral ventricular, left temporal lobe, and right putamen volumes. The immunological age, and immunosenescence in BD has been associated with an increased proportion of late differentiated T cells (CD3+CD8+CD28-CD27) in peripheral blood mononuclear cells (44 male and female euthymic patients with BD type I; β = 0.360, p = .013) [13]. Oxidative stress levels were higher in BD-mania for TBARS (P < .0001) and uric acid (P < .0001); in BD-depression for TBARS (P = .02); and BD-euthymia for uric acid (P = .03) and telomere shortening in a meta-analysis with forty-four studies (n = 3,767: BD = 1,979; HCs = 1,788) [14]. In addition, these patients more frequently present other medical conditions during the illness, such as cardiovascular and endocrine diseases and even dementia, leading some researchers to consider BD as a disorder of accelerated aging [15,16].”

2. You wrote: “Moreover, epidemiological studies have shown a correlation between telomere shortening and increased mortality due to cardiovascular and infection-related diseases (1,17). Subsequently, leukocyte telomere length (LTL) was associated with cellular senescence and longevity, as well as with disorders associated with aging (18). In addition to the variation in LTL with age, studies have recently reported heritability as a very important contributory factor to the variation in human LTL (19). A meta-analysis of telomere length studies showed high heritability for the LTL phenotype, estimated between 34-82% (20).” [Again, can you please present sample sizes, study design (prospective…cohorts) attrition, and numerical and statistical outcomes?] As written we have no idea of the risk of bias in the sample or strength of association].

The following changes were added to the text, which is underlined for ease of reading, on page 4-5:

“Moreover, a meta-analysis with twenty-four studies (43,725 participants) indicated that shorter telomere length is associated with risk of cardiovascular disease [19]. Furthermore, shorter telomere length was associated with infection-related diseases in 75,309 individuals randomly invited from Danish Civil Registration System (95% confidence interval) [20]. Subsequently, leukocyte telomere length (LTL) was associated with cellular senescence and longevity, as well as with disorders associated with aging [21]. In addition to the variation in LTL with age (aged 19-64 years at baseline and follow-up of 12 years), studies have recently reported heritability as a very important contributory factor to the variation in human LTL estimated at 64% (95% CI 39% to 83%) with 22% (95% CI 6% to 49%) of shared environmental effects in 355 monozygotic and 297 dizygotic (same-sex twins). Heritability of age-dependent LTL attrition rate was estimated at 28% (95% CI 16% to 44%) [21]. A meta-analysis of telomere length studies, with a total of 19,713 participants, showed high heritability for the LTL phenotype, estimated between 34-82% (95% CI 0.64–0.76) [22]. Another meta-analysis investigation, including 14,827 participants, reported LTL shortening in posttraumatic stress disorder (PTSD), anxiety disorders, depressive disorders, BD, and psychosis (Hedge’s g = −0.50, p< 0.001) [23]. With regard specifically to BD, both LTL reduction (22,23), and longer LTL [24]have been observed. However, the longer LTL is interpreted as an effect of lithium acting as a neuroprotective drug during the treatment of BD type 1 or 2 (n=256) and healthy controls (n=139), and BD had 35% longer telomeres compared with controls (P<0.0005) [24,25] . In another study 200 patients with BD had longer LTL, positively correlated with lithium treatment in patients treated for more than two years (p=0.037) [25]”. 

3. You wrote: “The biological function of telomeric length in leukocytes of BD patients is little explored in the literature,” [Please explain why you did not study telomere length in other cell types. Clearly leucocytes are related to your inflammation hypothesis. Can you please be explicit?

The following sentence was added in the text to justify the choice of cells used in the study, on page 5:

 “Leukocyte telomere length is therefore considered a promising biomarker of biological aging and accelerated aging and leukocytes had the same rate compared with skin cells, fat cells, and muscle tissues in a cohort of 87 subjects, being a peripheral blood sample more reliable with the rest of the human body [26].”

4. Sample: You wrote” This was a cross-sectional study using 22 families with two or more individuals per family affected by BD” [how many BD patients are there in your medical system? How were these patients chosen? (for another study?) How does this affect generalisability?]

We add the total number of individuals and the number of individuals with BD. We do not have the total number of this court for which a collaborative study was carried out and the families that passed the inclusion criteria were sent to us. The following changes were added to the text, which is underlined for ease of reading, on page 5:

 “This was a cross-sectional study using 22 families (143 individuals) with two or more individuals per family affected by BD totaling 60 individuals diagnosed with BD. DNA samples of these Family members are stored in a DNA bank from the Instituto de Psiquiatria do Hospital das Clínicas da Faculdade de Medicina da Universidade de São Paulo.”

5. Results: You wrote: “major depressive disorder (MDD), minor depression, schizophrenia, intermittent depressive disorder, alcoholism, hypomania, and dementia,” [How do the comorbidities affect your results?]

Thank you for this observation

In table 1 we describe that major depressive disorder (MDD) represents 11.9% of the BD population, minor depression and schizophrenia 3.5%, intermittent depressive disorder 2.8% and alcoholism, hypomania, and dementia less than 4% of populations in this study.

These results did not show statistical significance because they represent a very small percentage of the samples, and these comorbidities were expected for the BD Diagnosis. We describe in the discussion, pages 11 and 12, that some of these comorbidities may favor changes in telomeric length in a secondary way.

6. You wrote: The same standard curve of pooled DNA from these patient samples ranging from 10 ng to 0.016 ng was run on each plate for both genes and used to determine the quantity of each gene for each sample. This allowed control of the differences in the efficiencies between the Telomere and HBB.” [how did the differences in DNA sample size affect analysis reliability?]

The DNA standard curve from 10 ng to 0.016 ng, as described in the text, was performed according to the Cawthom protocol. The LTL quantification method calculates telomere and single copy gene abundance against a standard curve with different concentration points.

7. You wrote; “Figure 3 presents telomere length measurements according to severity of suicidal ideation (no periods, at least one week, two or four weeks of suicidal ideation). There was a significant difference between the groups (p = 0.02) and the estimate of the effect associated with suicidal ideation (β = - 0.06) indicates that the longer the period of suicidal ideation, the shorter the telomere length.”[This is an important outcome and is an important criterion in measuring strength of association according to Hill’s criteria].

Thanks for the comment and we agree with the importance of this result considering the values of B and the p value.

We thank you in advance for your consideration of our manuscript and look forward to hearing from you in due course.

Yours sincerely,

Looking forward to your evaluation, 

Homero Vallada, MD, PhD

---

## [Decision Letter · Decision Letter 1]

27 Sep 2022

Shorter telomere length and suicidal ideation in familial bipolar disorder

PONE-D-22-12118R1

Dear Dr. Vallada,

We’re pleased to inform you that your manuscript has been judged scientifically suitable for publication and will be formally accepted for publication once it meets all outstanding technical requirements.

Kind regards,

Vincenzo De Luca

Academic Editor

PLOS ONE

Additional Editor Comments (optional):

Reviewers' comments:

Reviewer's Responses to Questions

**Comments to the Author**

1. If the authors have adequately addressed your comments raised in a previous round of review and you feel that this manuscript is now acceptable for publication, you may indicate that here to bypass the “Comments to the Author” section, enter your conflict of interest statement in the “Confidential to Editor” section, and submit your "Accept" recommendation.

Reviewer #1: All comments have been addressed

2. Is the manuscript technically sound, and do the data support the conclusions?

Reviewer #1: Yes

3. Has the statistical analysis been performed appropriately and rigorously? 

Reviewer #1: Yes

4. Have the authors made all data underlying the findings in their manuscript fully available?

Reviewer #1: Yes

5. Is the manuscript presented in an intelligible fashion and written in standard English?

Reviewer #1: Yes

6. Review Comments to the Author

Reviewer #1: Thanks to the authors for their careful updating of the manuscript. The literature review is now evidence-based and the presentation of results appropriate for the available data.

7. PLOS authors have the option to publish the peer review history of their article (what does this mean?). If published, this will include your full peer review and any attached files.

Reviewer #1: **Yes: **Roger E. Thomas

---

## [Editor Report · Acceptance letter]

8 Nov 2022

PONE-D-22-12118R1 

Shorter telomere length and suicidal ideation in familial bipolar disorder 

Dear Dr. Vallada:

I'm pleased to inform you that your manuscript has been deemed suitable for publication in PLOS ONE. Congratulations! Your manuscript is now with our production department. 

Kind regards, 

on behalf of

Dr. Vincenzo De Luca 

Academic Editor

PLOS ONE